# LAPLACIAN SMOOTHING GRADIENT DESCENT

## ABSTRACT

We propose a class of very simple modifications of gradient descent and stochastic gradient descent. We show that when applied to a large variety of machine learning problems, ranging from softmax regression to deep neural nets, the proposed surrogates can dramatically reduce the variance and improve the generalization accuracy. The methods only involve multiplying the usual (stochastic) gradient by the inverse of a positive definitive matrix coming from the discrete Laplacian or its high order generalizations. The theory of Hamilton-Jacobi partial differential equations demonstrates that the implicit version of new algorithm is almost the same as doing gradient descent on a new function which (i) has the same global minima as the original function and (ii) is "more convex". We show that optimization algorithms with these surrogates converge uniformly in the discrete Sobolev $H_\sigma^p$ sense and reduce the optimality gap for convex optimization problems. We implement our algorithm into both PyTorch and Tensorflow platforms which only involves changing of a few lines of code. The code will be available on Github.

## 1 INTRODUCTION

Stochastic gradient descent (SGD) has been the workhorse for solving large-scale machine learning problems (Bottou et al., 2018). It gives rise to a family of algorithms that make training of deep neural nets (DNN) practical, which is believed to somehow implicitly smooth the loss function of the DNN (Jastrzebski et al., 2018). Many efforts have been carried out to improve training and generalization of DNN by directly searching for flat minima (Keskar et al., 2017; Chaudhari et al., 2017; 2016). An alternative view of SGD's magic comes from the theory of uniform stability (Bousquet & Elisseeff, 2002; Duchi et al., 2011; Hardt et al., 2016; Bottou et al., 2016; Gonen & Shalev-Shwartz, 2017).

The noise in SGD, on the one hand, helps gradient-based optimization algorithms circumvent spurious local minima and reach those that generalize well (Schmidhuber, 2014). On the other hand, it slows down the convergence of regular gradient descent (GD). To recover the linear convergence rate for strongly convex functions, several interesting variance reduction algorithms have been proposed, e.g., SAGA (Defazio & Bach, 2014) and SVRG (Johoson & Zhang, 2013). These algorithms have a certain amount of difficulty in training DNN. SAGA has a relatively high space complexity in storing the gradient for many samples. SVRG requires computation of the full batch gradient.

In this work, we propose a carefully designed positive definite matrix to smooth and to reduce variance of the (stochastic) gradient on-the-fly. The resulting surrogate tends to reduce noise in SGD and improve training of DNN. We call this procedure Laplacian smoothing. The gradient smoothing can be done by multiplying the gradient by the inverse of the following circulant convolution matrix

$$
\boldsymbol{A}_\sigma := \begin{bmatrix} 1+2\sigma & -\sigma & 0 & \dots & 0 & -\sigma \\ -\sigma & 1+2\sigma & -\sigma & \dots & 0 & 0 \\ 0 & -\sigma & 1+2\sigma & \dots & 0 & 0 \\ \dots & \dots & \dots & \dots & \dots & \dots \\ -\sigma & 0 & 0 & \dots & -\sigma & 1+2\sigma \end{bmatrix} \tag{1}
$$

for some positive constant $\sigma \geq 0$. In fact, we can write $\boldsymbol{A}_\sigma = \boldsymbol{I} - \sigma\boldsymbol{L}$, where $\boldsymbol{I}$ is the identity matrix, and $\boldsymbol{L}$ is the discrete one-dimensional Laplacian which acts on indices. We define the (periodic)

forward finite difference matrix as

$$
\boldsymbol{D}_+ = \begin{bmatrix} -1 & 1 & 0 & \ldots & 0 & 0 \\ 0 & -1 & 1 & \ldots & 0 & 0 \\ 0 & 0 & -1 & \ldots & 0 & 0 \\ \ldots & \ldots & \ldots & \ldots & \ldots & \ldots \\ 1 & 0 & 0 & \ldots & 0 & -1 \end{bmatrix}.
$$

Then, we have $\boldsymbol{A}_\sigma = \boldsymbol{I} - \sigma \boldsymbol{D}_- \boldsymbol{D}_+$, where $\boldsymbol{D}_- = -\boldsymbol{D}_+^\top$ is the backward finite difference. The resulting Laplacian smoothing stochastic gradient descent (LS-SGD) requires negligible extra computational cost and generalizes better than the standard SGD. When the Hessian has a poor condition number, gradient descent performs poorly. In this case, the derivative increases rapidly in one direction, while increasing slowly in others. Gradient smoothing can avoid jitter along steep directions and help make progress in shallow directions (Li & et al, 2018). Moreover, we show that the operator $\boldsymbol{A}_\sigma^{-1}$ plays role as a denoiser which enables better convergence in the presence of a very noisy stochastic gradient. The implicit version of our proposed approach is linked to an unusual Hamilton-Jacobi partial differential equation (HJ-PDE) whose solution makes the original loss function more convex while retaining its flat (and global) minima, and essentially works on this surrogate function with a much better landscape.

## 2 HAMILTON-JACOBI PDEs AND CONVEXIFICATION

Machine learning problems are generally formulated as finding the optimal parameters $\boldsymbol{w}$ of a parametric function $\boldsymbol{y} = h(\boldsymbol{x}, \boldsymbol{w})$, such that for an input $\boldsymbol{x}$, the output $\boldsymbol{y}$ is close to the ground-truth. The optimal $\boldsymbol{w}$ can be obtained by minimizing an empirical risk function, $f(X, Y, \boldsymbol{w}) \doteq f(\boldsymbol{w})$, given the training data $\{X, Y\}$. We start from the following unusual HJ-PDE with $f(\boldsymbol{w})$ as initial condition

$$
\begin{cases} u_t + \frac{1}{2}\langle \nabla_{\boldsymbol{w}} u, \boldsymbol{A}_\sigma^{-1} \nabla_{\boldsymbol{w}} u \rangle = 0, & (\boldsymbol{w}, t) \in \Omega \times [0, \infty) \\ u(\boldsymbol{w}, 0) = f(\boldsymbol{w}), & \boldsymbol{w} \in \Omega \end{cases} \tag{2}
$$

By the Hopf-Lax formula (Evans, 2010), the unique viscosity solution to Eq. (2) is represented by

$$
u(\boldsymbol{w}, t) = \inf_{\boldsymbol{v}} \left\{ f(\boldsymbol{v}) + \frac{1}{2t}\langle \boldsymbol{v} - \boldsymbol{w}, \boldsymbol{A}_\sigma(\boldsymbol{v} - \boldsymbol{w}) \rangle \right\}.
$$

This viscosity solution $u(\boldsymbol{w}, t)$ makes $f(\boldsymbol{w})$ "more convex", an intuitive definition and theoretical explanation of "more convex" can be found in (Chaudhari et al., 2017; 2016), by bringing down the local maxima while retaining and widening local minima. An illustration of this is shown in Fig. 1. If we perform the smoothing GD with proper step size on the function $u(\boldsymbol{w}, t)$, it is easier to reach the global or at least a flat minima of the original nonconvex function $f(\boldsymbol{w})$.

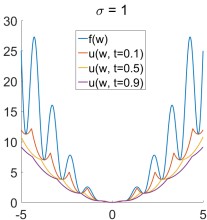

Figure 1: $f(\boldsymbol{w}) = \|\boldsymbol{w}\|^2 \left(1 + \frac{1}{2}\sin(2\pi\|\boldsymbol{w}\|)\right)$ is made more convex by solving Eq.(2). The plot shows the cross section of the 5D problem with $\sigma = 1$ and different $t$ values.

**Proposition 1.** *Suppose $f(\boldsymbol{w})$ is differentiable, the LS-GD on $u(\boldsymbol{w}, t)$*

$$
\boldsymbol{w}^{k+1} = \boldsymbol{w}^k - t\boldsymbol{A}_\sigma^{-1}\nabla_{\boldsymbol{w}} u(\boldsymbol{w}^k, t)
$$

*is equivalent to the smoothing implicit GD on $f(\boldsymbol{w})$*

$$
\boldsymbol{w}^{k+1} = \boldsymbol{w}^k - t\boldsymbol{A}_\sigma^{-1}\nabla f(\boldsymbol{w}^{k+1}). \tag{3}
$$

All the proofs here and below are provided in the appendix.

## 2.1 LAPLACIAN SMOOTHING GRADIENT DESCENT

Laplacian smoothing implicit gradient descent requires inner iterations as used in (Chaudhari et al., 2017), which is computationally expensive. We consider the following explicit scheme

$$\boldsymbol{w}^{k+1} = \boldsymbol{w}^k - \gamma_k \boldsymbol{A}_\sigma^{-1} \nabla f(\boldsymbol{w}^k).$$

Intuitively, compared to the standard GD, this scheme smooths the gradient on-the-fly by an elliptic smoothing operator. We adopt fast Fourier transform (FFT) to compute $\boldsymbol{A}_\sigma^{-1} \nabla f(\boldsymbol{w}^k)$, which is available in both PyTorch (Paszke et al., 2017) and TensorFlow (Abadi et al., 2016). Given a vector $\boldsymbol{g}$, a smoothed vector $\boldsymbol{d}$ can be obtained by computing $\boldsymbol{d} = \boldsymbol{A}_\sigma^{-1} \boldsymbol{g}$. This is equivalent to $\boldsymbol{g} = \boldsymbol{d} - \sigma \boldsymbol{v} * \boldsymbol{d}$, where $\boldsymbol{v} = [-2, 1, 0, \cdots, 0, 1]^\top$ and $*$ is the convolution operator. Therefore

$$\boldsymbol{d} = \text{ifft}\left(\frac{\text{fft}(\boldsymbol{g})}{\mathbf{1} - \sigma \cdot \text{fft}(\boldsymbol{v})}\right),$$

where we use component-wise division, fft and ifft are the FFT and inverse FFT, respectively. Hence, the gradient smoothing can be done in quasilinear time. This additional time complexity is almost the same as performing a one step update on the weights vector $\mathbf{w}$. For many machine learning models, we may need to concatenate the parameters into a vector. This reshape might lead to some ambiguity, nevertheless, based on our tests, both row and column majored reshaping work for the LS-GD algorithm. Moreover, in deep learning cases, the weights in different layers might have different physical meanings. We then perform layer-wise gradient smoothing, instead.

**Remark 1.** *In image processing, the Sobolev gradient (Jung et al., 2009) involves a multi-dimensional Laplacian operator which operates on $\boldsymbol{w}$, is different from the one-dimensional discrete Laplacian operator employed in our LS-GD scheme, which operates on indices.*

We first show that LS-GD can help bypass sharp minima and reach the global minima. We consider the following function, in which we 'drill' narrow holes on a smooth convex function,

$$f(x,y,z) = -4e^{-\left((x-\pi)^2+(y-\pi)^2+(z-\pi)^2\right)} - 4\sum_i \cos(x)\cos(y)e^{-\beta\left((x-r\sin(\frac{i}{2})-\pi)^2+(y-r\cos(\frac{i}{2})-\pi)^2\right)}, \quad (4)$$

where the summation is taken over the index set $\{i \in \mathbb{N}| \ 0 \leq i < 4\pi\}$, $r$ and $\beta$ are the parameters that determine the location and narrowness of the local minima and are set to $1$ and $\frac{1}{\sqrt{500}}$, respectively. We do GD and LS-GD starting from a random point in the neighborhoods of the narrow minima, i.e., $(x_0, y_0, z_0) \in \{\bigcup_i U_\delta(r\sin(\frac{i}{2}) + \pi, r\cos(\frac{i}{2}) + \pi)| \ 0 \leq i < 4\pi, i \in \mathbb{N}\}$, where $U_\delta(P)$ is a neighborhood of the point $P$ with radius $\delta$. Our experiments (Fig. 2) show that, if $\delta \leq 0.2$, GD will converge to narrow local minima, while LS-GD convergences to wider global minima.

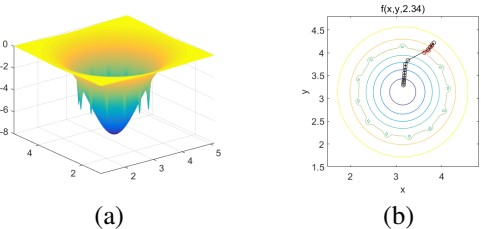

(a)            (b)

Figure 2: Demo of GD and LS-GD. Panel (a) depicts the slice of the function (Eq.(4)) with $z = 2.34$; panel (b) shows the paths of GD (red) and LS-GD (black). We take the step size to be 0.02 for both GD and LS-GD. $\sigma = 1.0$ is utilized for LS-GD.

## 2.2 GENERALIZED SMOOTHING GRADIENT DESCENT

We can generalize $\boldsymbol{A}_\sigma$ to the $n$th order discrete hyper-diffusion operator as follows

$$\boldsymbol{I} + (-1)^n \sigma \boldsymbol{L}^n \doteq \boldsymbol{A}_\sigma^n.$$

Each row of the discrete Laplacian operator $\boldsymbol{L}$ consists of an appropriate arrangement of weights in central finite difference approximation to the 2nd order derivative. Similarly, each row of $\boldsymbol{L}^n$ is an arrangement of the weights of the central finite difference to approximate the $2n$th order derivative.

**Remark 2.** *The $n$th order smoothing operator $\boldsymbol{I} + (-1)^n \sigma \boldsymbol{L}^n$ can only be applied to the problem with dimension at least $2n + 1$. Otherwise, we need to add dummy variables to the object function.*

Again, we apply FFT to compute the smoothed gradient vector. For a given gradient vector $\boldsymbol{g}$, the smoothed surrogate, $(\boldsymbol{A}_\sigma^n)^{-1}\boldsymbol{g} \doteq \boldsymbol{d}$, can be obtained by solving $\boldsymbol{g} = \boldsymbol{d} + (-1)^n \sigma \boldsymbol{v}_n * \boldsymbol{d}$, where $\boldsymbol{v}_n = (c_{n+1}^n, c_{n+2}^n, \cdots, c_{2n+1}^n, 0, \cdots, 0, c_1^n, c_2^n, \cdots, c_{n-1}^n, c_n^n)$ is a vector of the same dimension as the gradient to be smoothed. And the coefficient vector $\boldsymbol{c}^n = (c_1^n, c_2^n, \cdots, c_{2n+1}^n)$ can be obtained recursively by the following formula

$$\boldsymbol{c}^1 = (1, -2, 1), \quad c_i^n = \begin{cases} 1 & i = 1, 2n+1 \\ -2c_1^{n-1} + c_2^{n-1} & i = 2, 2n \\ c_{i-1}^{n-1} - 2c_i^{n-1} + c_{i+1}^{n-1} & \text{otherwise.} \end{cases}$$

**Remark 3.** *The computational complexities for different order smoothing schemes are the same when the FFT is utilized for computing the surrogate gradient.*

## 3    REDUCE OPTIMALITY GAP IN SGD

We show advantages of the LS-(S)GD and generalized schemes for convex optimization. Consider finding the minima $\boldsymbol{x}^*$ of the quadratic function $f(\boldsymbol{x})$ defined in Eq. (5) by different schemes.

$$f(x_1, x_2, \cdots, x_{100}) = \sum_{i=1}^{50} x_{2i-1}^2 + \sum_{i=1}^{50} \frac{x_{2i}^2}{10^2}. \tag{5}$$

To simulate SGD, we add Gaussian noise to the gradient vector, i.e., at a given point $\boldsymbol{x}$, we have

$$\tilde{\nabla}_\epsilon f(\boldsymbol{x}) := \nabla f(\boldsymbol{x}) + \epsilon \mathcal{N}(\boldsymbol{0}, \boldsymbol{I}),$$

where the scalar $\epsilon$ controls the noise level, $\mathcal{N}(\boldsymbol{0}, \boldsymbol{I})$ is the vector with zero mean and unit variance in each coordinate. The corresponding numerical schemes can be formulated as

$$\boldsymbol{x}^{k+1} = \boldsymbol{x}^k - \eta_k (\boldsymbol{A}_\sigma^n)^{-1} \tilde{\nabla}_\epsilon f(\boldsymbol{x}^k), \tag{6}$$

where $\sigma$ is the smoothing parameter selected to be 10.0 to kill the intense noise. We take diminishing step sizes with initial values 0.1 for SGD/smoothed SGD; 0.9 and 1.8 for GD/smoothed GD, respectively. Without noise, the smoothing allows us to take larger step sizes, rounding to the first digit, 0.9 and 1.9 are the largest suitable step size for GD and smoothed version here. We compare constant learning rate and exponentially decaying learning rate, i.e., after every 1000 iteration, the learning rate is divided by 10. We apply different schemes that corresponding to $n = 0, 1, 2$ in Eq. (6) to the problem Eq. (5), with the initial point $\boldsymbol{x}^0 = (1, 1, \cdots, 1)$.

Figure. 3 shows the iteration v.s. optimality gap when the constant learning rate is applied to different noise levels. In the noise free case, all three schemes converge linearly, but gradient smoothing has a smaller decay constant due to its increased condition number. When there is noise, our smoothed gradient helps to reduce the optimality gap and converges faster after a few iterations.

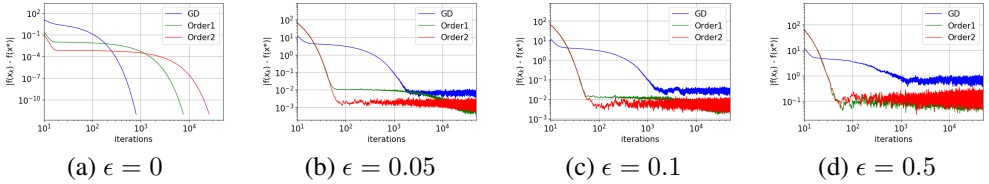

(a) $\epsilon = 0$        (b) $\epsilon = 0.05$        (c) $\epsilon = 0.1$        (d) $\epsilon = 0.5$

Figure 3: Iterations v.s. optimality gap for GD and smoothed GD with order 1 and 2 for the problem in Eq.(5). Constant step size was used.

The exponentially decaying learning rate helps our smoothed SGD to reach a point with a smaller optimality gap, and the higher order smoothing further reduce the optimality gap, as shown in Fig. 4. One simple reason for this in the noisy case is because of the noise removal properties of the smoothing operators. The influence of the learning rate is still under investigation. We establish the convergence of our proposed smoothing gradient descent algorithms.

### 3.1 Some Properties of Laplacian Smoothing Gradient Descent

We say the objective function $f$ has $L$-Lipschitz gradient, if for any $\boldsymbol{w}, \boldsymbol{u} \in \mathbb{R}^m$, we have $\|\nabla f(\boldsymbol{w}) - \nabla f(\boldsymbol{u})\| \leq L\|\boldsymbol{w} - \boldsymbol{u}\|$, and $f$ is $a$-strongly convex, if $\langle \nabla f(\boldsymbol{w}) - \nabla f(\boldsymbol{u}), \boldsymbol{w} - \boldsymbol{u} \rangle \geq a\|\boldsymbol{w} - \boldsymbol{u}\|^2$. We define the vector norm induced by any matrix $\boldsymbol{A}$ as $\|\boldsymbol{w}\|_{\boldsymbol{A}} := \sqrt{\langle \boldsymbol{w}, \boldsymbol{A}\boldsymbol{w} \rangle}$.

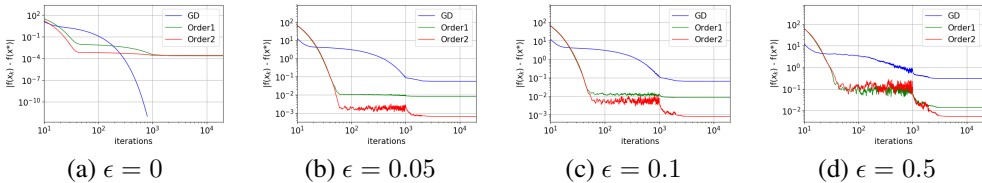

(a) $\epsilon = 0$     (b) $\epsilon = 0.05$     (c) $\epsilon = 0.1$     (d) $\epsilon = 0.5$

Figure 4: Iterations v.s. optimality gap for GD and smoothed GD with order 1 and 2 for the problem in Eq.(5). Exponentially decaying step size is utilized here.

**Proposition 2.** *Suppose $f$ is convex with the global minimizer $\boldsymbol{w}^*$, and $f^* = f(\boldsymbol{w}^*)$. Consider the following iteration with constant learning rate $\eta > 0$*

$$\boldsymbol{w}^{k+1} = \boldsymbol{w}^k - \eta(\boldsymbol{A}_\sigma^n)^{-1}\boldsymbol{g}^k,$$

*where $\boldsymbol{g}^k$ is the sampled gradient in the $k$th iteration at $\boldsymbol{w}^k$ satisfying $\mathbb{E}[\boldsymbol{g}^k] = \nabla f(\boldsymbol{w}^k)$. Denote $G_{\boldsymbol{A}_\sigma^n} := \lim_{K \to \infty} \frac{1}{K} \sum_{k=0}^{K-1} \|\boldsymbol{g}^k\|_{(\boldsymbol{A}_\sigma^n)^{-1}}^2$ and $\overline{\boldsymbol{w}}^K := \sum_{k=0}^{K-1} \boldsymbol{w}^k / K$ the ergodic average of iterates. Then the optimality gap is*

$$\lim_{K \to \infty} \mathbb{E}[f(\overline{\boldsymbol{w}}^K)] - f^* \leq \frac{\eta G_{\boldsymbol{A}_\sigma^n}}{2}.$$

Note that $\|\boldsymbol{g}\|_{(\boldsymbol{A}_\sigma^n)^{-1}}$ generally decreases in $n$ unless $\boldsymbol{g}$ is constant, which indicates that a bigger $n$ implies smaller optimality gap. This is consistent with the experimental results above.

**Proposition 3.** *Suppose $f$ is $L$-Lipschitz smooth and $a$-strongly convex with the global minimizer $\boldsymbol{w}^*$. Consider the generalized smoothing gradient descent algorithm*

$$\boldsymbol{w}^{k+1} = \boldsymbol{w}^k - \eta_k(\boldsymbol{A}_\sigma^n)^{-1}\boldsymbol{g}^k,$$

*where $\boldsymbol{g}^k$ is the sampled gradient in the $k$th iteration at $\boldsymbol{w}^k$ satisfying $\mathbb{E}\left[\boldsymbol{g}^k\right] = \nabla f(\boldsymbol{w}^k)$ and $\mathbb{E}\left[\|\boldsymbol{g}^k\|_{(\boldsymbol{A}_\sigma^n)^{-1}}^2\right] \leq C_0 + C_1\|\nabla f(\boldsymbol{w}^k)\|^2$ for all $k \in \mathbb{N}$. If we take $\eta_k = \frac{C}{k+1}$ for some $C > 0$, then we have*

$$\mathbb{E}\left[\|\boldsymbol{w}^k - \boldsymbol{w}^*\|_{\boldsymbol{A}_\sigma^n}^2\right] = \mathbb{E}\left[\|\boldsymbol{w}^k - \boldsymbol{w}^*\|^2 + \sigma\|\boldsymbol{D}_+^n(\boldsymbol{w}^k - \boldsymbol{w}^*)\|^2\right] = O\left(\frac{1}{k+1}\right),$$

*i.e., we have $H_\sigma^n$ uniform convergence in $\sigma$ of $\{\boldsymbol{w}^k\}$ in expectation. The $H_\sigma^n$ norm of $\boldsymbol{w}$ is defined by $\|\boldsymbol{w}\|_\sigma^n := \|w\|_{\boldsymbol{A}_\sigma^n} = \sqrt{\langle \boldsymbol{w}, \boldsymbol{A}_\sigma^n \boldsymbol{w} \rangle}$.*

**Proposition 4.** *Consider the algorithm $\boldsymbol{w}^{k+1} = \boldsymbol{w}^k - \eta_k(\boldsymbol{A}_\sigma^n)^{-1}\nabla f(\boldsymbol{w}^k)$. Suppose $f$ is convex and $L$-Lipschitz smooth. If the step size satisfies $0 < \underline{\eta} \leq \eta \leq \overline{\eta} < \frac{2}{L}$. Then $\lim_{t \to \infty} \|\nabla f(\boldsymbol{w}^k)\| \to 0$. Moreover, if the Hessian $\nabla^2 f$ of $f$ is continuous with $\boldsymbol{w}^*$ being the global minimizer of $f$, and $\overline{\eta}\|\nabla^2 f\| < 1$, then $\|\boldsymbol{w}^k - \boldsymbol{w}^*\|_{\boldsymbol{A}_\sigma^n} \to 0$ as $k \to \infty$, and the convergence is linear and independent of $\sigma$.*

In what follows, we present the noise reduction properties of the proposed smoothing operator $\boldsymbol{A}_\sigma^{-1}$.

**Proposition 5.** *For any vector $\boldsymbol{g} \in \mathbb{R}^m$, $\boldsymbol{d} = \boldsymbol{A}_\sigma^{-1}\boldsymbol{g}$, let $j_{\max} = \arg\max_i d_i$ and $j_{\min} = \arg\min_i d_i$. We have $\max_i d_i = d_{j_{\max}} \leq g_{j_{\max}} \leq \max_i g_i$ and $\min_i d_i = d_{j_{\min}} \geq g_{j_{\min}} \geq \min_i g_i$.*

**Proposition 6.** *The operator $\boldsymbol{A}_\sigma^{-1}$ preserves the sum of components. For any $\boldsymbol{g} \in \mathbb{R}^m$ and $\boldsymbol{d} = \boldsymbol{A}_\sigma^{-1}\boldsymbol{g}$, we have $\sum_j d_j = \sum_j g_j$, or equivalently, $\boldsymbol{1}^\top \boldsymbol{d} = \boldsymbol{1}^\top \boldsymbol{g}$.*

**Proposition 7.** *Given any vector $\boldsymbol{g} \in \mathbb{R}^m$ and $\boldsymbol{d} = \boldsymbol{A}_\sigma^{-1}\boldsymbol{g}$, then*

$$\|\boldsymbol{d}\| + \sigma \frac{\|\boldsymbol{D}_+\boldsymbol{d}\|^2}{\|\boldsymbol{d}\|} \leq \|\boldsymbol{g}\|.$$

*The above inequality is strict unless $\boldsymbol{g} = \boldsymbol{d}$ is a constant vector. In particular, we have $\|\boldsymbol{d}\| \leq \|\boldsymbol{g}\|$ and $\|\boldsymbol{D}_+\boldsymbol{d}\| \leq \frac{1}{\sqrt{\sigma}}\|\boldsymbol{g}\|$.*

Let $g$ be the noise vector contained in the stochastic gradient, the above results imply that the extreme values in $A_\sigma^{-1}g$ are smaller than those in $g$ (in magnitude), and it also has a much smaller $\ell_2$ norm.

**Proposition 8.** *For any* $g \in \mathbb{R}^m$, *define* $\mathrm{Var}(g) := \frac{1}{m}\|g\|^2 - \left(\frac{\mathbf{1}^\top g}{m}\right)^2$ *be the variance of components in* $g$. *Let* $d = A_\sigma^{-1}g$, *then*

$$\mathrm{Var}(d) \leq \mathrm{Var}(g) - 2\sigma\frac{\|D_+d\|^2}{m} - \sigma^2\frac{\|D_+d\|^4}{m\|d\|^2}.$$

*The inequality is strict unless* $g = d$ *is a constant vector.*

Proposition 8 shows that the component-wise variance of $A_\sigma^{-1}g$ is considerably less than that of $g$, unless $g$ is a constant vector. Our last result shows that $A_\sigma^{-1}g$ has diminishing $\ell_1$ norm of finite difference of all orders. This is an excellent desnoising result.

**Proposition 9.** *Given vectors* $g$ *and* $d = A_\sigma^{-1}g$, *for any* $p \in \mathbb{N}$, *it holds that* $\|D_+^p d\|_1 \leq \|D_+^p g\|_1$. *The inequality is strict unless* $D_+^p g$ *is a constant vector.*

**Remark 4.** *The above proofs generalize for* $n > 1$, *except for Propositions 5 and 9.*

### 3.2 SOFTMAX REGRESSION

Consider applying the proposed optimization schemes to Softmax regression. We run 200 epochs of SGD and different order smoothing algorithms to maximize the likelihood of Softmax regression with batch size 100. Based on the results from previous section, we apply the exponentially decay learning rate with initial value 0.1 and decay 10 times after every 50 epochs. We train the model with only 10 % randomly selected MNIST training data and test the trained model on the entire testing images. We further compare with SVRG under the same setting. Figure. 5 shows the histograms of generalization accuracy of Softmax regression model trained by SGD ((a)); SVRG ((b)); LS-SGD (order 1) ((c)); LS-SGD (oder 2) ((d)). It is seen that SVRG improves the generalization with higher average accuracy. But the first and second order smoothing schemes significantly improve averaged generalization accuracy by more than 1% and reduce the variance over 100 independent trials. The training loss of these 100 experiments by different optimization algorithms are shown in the appendix.

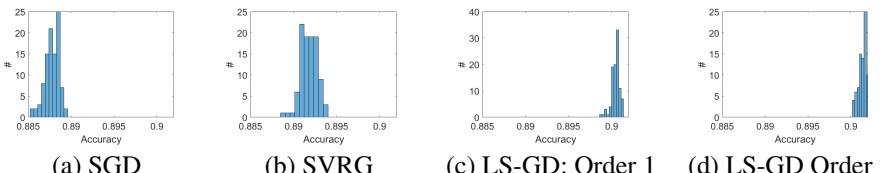

| (a) SGD | (b) SVRG | (c) LS-GD: Order 1 | (d) LS-GD Order 2 |

Figure 5: Testing accuracy of Softmax model trained on randomly selected $10\%$ MNIST data.

## 4 APPLICATIONS TO DEEP NEURAL NETS

### 4.1 TRAIN NEURAL NETS WITH SMALL BATCH SIZE

Many advanced artificial intelligence tasks make high demand on training neural nets with extremely small batch size. The milestone technique for this is group normalization (Wu & He, 2018). In this section, we show that LS-SGD successfully trains DNN with extremely small batch size. We consider LeNet-5 devised by (LeCun et al., 1998) for MNIST classification. Our network architecture is as follows

$$\text{LeNet-5: input}_{28\times28} \to \text{conv}_{20,5,2} \to \text{conv}_{50,5,2} \to \text{fc}_{512} \to \text{softmax}.$$

The notation $\text{conv}_{c,k,m}$ denotes a 2D convolutional layer with $c$ output channels, each of which is the sum of a channel-wise convolution operation on the input using a learnable kernel of size $k \times k$, it further adds ReLU nonlinearity and max pooling with stride size $m$. $\text{fc}_{512}$ is an affine transformation that transforms the input to a vector of dimension 512. Finally, the tensors are activated by a softmax function. The MNIST data is first passed to the layer $\text{input}_{28\times28}$, and further processed by this

hierarchical structure. We run 100 epochs of both SGD and LS-SGD with initial learning rate 0.01 and divide by 5 after 50 epochs, and use a weight decay of 0.0001 and momentum of 0.9. Figure. 6(a) plots the generalization accuracy on the test set with the LeNet5 trained with different batch sizes. For each batch size, LS-SGD with $\sigma = 1.0$ keeps the testing accuracy more than $99.4\%$, SGD reduce the accuracy to $97\%$ when batch size 4 is used. The classification become just a random guess, when the model is trained by SGD with batch size 2. Small batch size leads to large noise in the gradient, which may make the noisy gradient not along the decent direction, However, Lapacian smoothing rescues this by killing the noise.

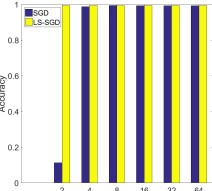 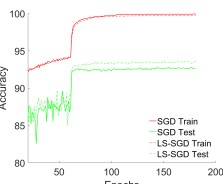

Figure 6: (a). Testing accuracy of LeNet5 trained by SGD/LS-SGD on MNIST with various batch sizes. (b). The evolution of the pre-activated ResNet56's training and generalization accuracy by SGD and LS-SGD. (Start from the 20-th epoch.)

## 4.2 IMPROVE GENERALIZATION ACCURACY

The skip connections in ResNet smooth the landscape of the loss function of the classical CNN (He et al., 2016; Li et al., 2017). This means that ResNet has fewer sharp minima. On Cifar10 (Krizhevsky, 2009), we compare the performance of LS-SGD and SGD on ResNet with the pre-activated ResNet56 as an illustration. We take the same training strategy as that used in (He et al., 2016), except that we run 200 epochs with the learning rate decaying by a factor of 5 after every 40 epochs. For ResNet, instead of applying LS-SGD for all epochs, we only use LS-SGD in the first 40 epochs, and the remaining training is carried out by SGD. The parameter $\sigma$ is set to $1.0$. Figure 6(b) depicts one path of the training and generalization accuracy of the neural nets trained by SGD and LS-SGD, respectively. It is seen that, even though the training accuracy obtained by SGD is higher than that by LS-SGD, the generalization is however inferior to that of LS-SGD. We conjecture that this is due to the fact that SGD gets trapped into some sharp but deeper minimum, which fits better than a flat minimum but generalizes worse. We carry out 25 replicas of this experiments, the histograms of the corresponding accuracy are shown in Fig. 7.

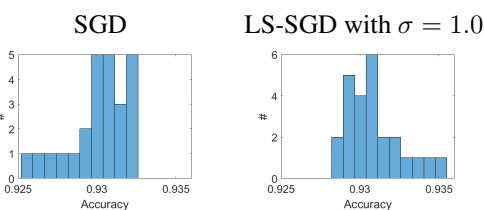

Figure 7: The histogram of the generalization accuracy of the pre-activated ResNet56 on Cifar10 trained with LS-SGD over 25 independent experiments.

## 4.3 TRAINING WASSERSTERIN GAN

Generative Adversarial Networks (GANs) (Goodfellow et al., 2014) are notoriously delicate and unstable to train (Arjovsky & Bottou, 2017). In (M. Arjovsky & Bottou, 2017), Wasserstein-GANs (WGANs) are introduced to combat the instability in the training GANs. In addition to being more robust in training parameters and network architecture, WGANs provide a reliable estimate of the Earth Mover (EM) metric which correlates well with the quality of the generated samples. Nonetheless, WGANs training becomes unstable with a large learning rate or when used with a momentum based optimizer (M. Arjovsky & Bottou, 2017). In this section, we demonstrate that the gradient smoothing technique in this paper alleviates the instability in the training, and improves the quality of generated samples. Since WGANs with weight clipping are typically trained with RMSProp

(Tieleman & Hinton, 2012), we propose replacing the gradient $g$ by a smoothed version $g_\sigma = A_\sigma^{-1} g$, and also update the running averages using $g_\sigma$ instead of $g$. We name this algorithm LS-RMSProp.

To accentuate the instability in training and demonstrate the effects of gradient smoothing, we deliberately use a large learning rate for training the generator. We compare the regular RMSProp with the LS-RMSProp. The learning rate for the critic is kept small and trained approximately to convergence so that the critic loss is still an effective approximation to the Wasserstein distance.To control the number of unknowns in the experiment and make a meaningful comparison using the critic loss, we use the classical RMSProp for the critic, and only apply LS-RMSProp to the generator.

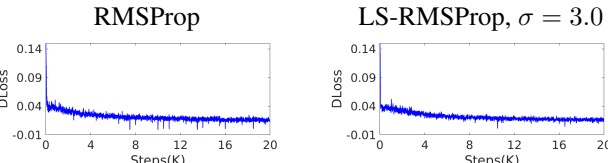

Figure 8: Critic loss with learning rate $lrD = 0.0001$, $lrG = 0.005$ for RMSProp (Left) and LS-RMSProp (Right), trained for 20K iterations. We apply a mean filter of window size 13 for better visualization. The loss from LS-RMSProp is visibly less noisy.

We train the WGANs on the MNIST dataset using the DCGAN (Radford et al., 2015) for both the critic and generator. In Figure 8 (left), we observe the loss for RMSProp trained with a large learning rate has multiple sharp spikes, indicating instability in the training process. The samples generated are also lower in quality, containing noisy spots as shown in Figure 9 (a). In contrast, the curve of training loss for LS-RMSProp is smoother and exhibits fewer spikes. The generated samples as shown in Fig. 9 (b) are also of better quality and visibly less noisy. The generated characters shown in Fig. 9 (b) are more realistic compared to the ones shown in Fig. 9 (a). The effects are less pronounced with a small learning rate, but still result in a modest improvement in sample quality as shown in Figure 9 (c) and (d).We also apply LS-RMSProp for training the critic, but do not see a clear improvement in the quality. This may be because the critic is already trained near optimality during each iteration, and does not benefit much from gradient smoothing.

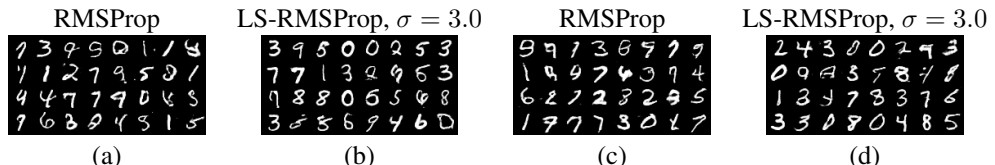

Figure 9: Samples from WGANs trained with RMSProp (a, c) and LS-RMSProp (b, d). The learning rate is set to $lrD = 0.0001$, $lrG = 0.005$ for both RMSProp and LS-RMSProp in (a) and (b). And $lrD = 0.0001$, $lrG = 0.0001$ are used for both RMSProp and LS-RMSProp in (c) and (d). The critic is trained for 5 iterations per step of the generator, and 200 iterations per every 500 steps of the generator.

## 4.4 DEEP REINFORCEMENT LEARNING

Finally, we apply the LS-SGD to deep reinforcement learning. We provide a detailed discussion and present the numerical result in the appendix.

## 5 CONCLUDING REMARKS

Motivated by the theory of Hamilton-Jacobi partial differential equations, we proposed Laplacian smoothing gradient descent and its high order generalizations. This simple modification dramatically reduces the optimality gap in stochastic gradient descent and helps to find better minima. Extensive numerical examples ranging from toy cases to shallow and deep neural nets to generative adversarial networks and to deep reinforcement learning, all demonstrate the advantage of the proposed smoothed gradient. Several issues remain, in particular devising an on-the-fly adaptive method for choosing the smoothing parameter $\sigma$ instead of using a fixed value.

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

## 6 APPENDIX

### 6.1 TECHNICAL PROOFS

**Proposition 1.** *Suppose $f(\boldsymbol{w})$ is differentiable, the Laplacian smoothing GD update on $u(\boldsymbol{w}, t)$*
$$\boldsymbol{w}^{k+1} = \boldsymbol{w}^k - t\boldsymbol{A}_\sigma^{-1}\nabla_{\boldsymbol{w}}u(\boldsymbol{w}^k, t)$$
*permits the smoothing implicit gradient descent on $f(\boldsymbol{w})$*
$$\boldsymbol{w}^{k+1} = \boldsymbol{w}^k - t\boldsymbol{A}_\sigma^{-1}\nabla f(\boldsymbol{w}^{k+1}).$$

**Proof of Proposition 1.** We define
$$z(\boldsymbol{w}, \boldsymbol{v}, t) := f(\boldsymbol{v}) + \frac{1}{2t}\langle \boldsymbol{v} - \boldsymbol{w}, \boldsymbol{A}_\sigma(\boldsymbol{v} - \boldsymbol{w})\rangle,$$
and rewrite $u(\boldsymbol{w}, t) = \inf_{\boldsymbol{v}} z(\boldsymbol{w}, \boldsymbol{v}, t)$ as $z(\boldsymbol{w}, \boldsymbol{v}(\boldsymbol{w}, t), t)$, where $\boldsymbol{v}(\boldsymbol{w}, t) = \arg\min_{\boldsymbol{v}} z(\boldsymbol{w}, \boldsymbol{v}, t)$. Then by the Euler-Lagrange equation,
$$\nabla_{\boldsymbol{w}}u(\boldsymbol{w}, t) = \nabla_{\boldsymbol{w}}z(\boldsymbol{w}, \boldsymbol{v}(\boldsymbol{w}, t), t) = \boldsymbol{J}_{\boldsymbol{w}}\boldsymbol{v}(\boldsymbol{w}, t)\nabla_{\boldsymbol{v}}z(\boldsymbol{w}, \boldsymbol{v}(\boldsymbol{w}, t), t) + \nabla_{\mathbf{w}}z(\boldsymbol{w}, \boldsymbol{v}(\boldsymbol{w}, t), t),$$
where $\boldsymbol{J}_{\boldsymbol{w}}\mathbf{v}(\boldsymbol{w}, t)$ is the Jacobian matrix of $\boldsymbol{v}$ w.r.t. $\boldsymbol{w}$. Notice that $\nabla_{\boldsymbol{v}}z(\boldsymbol{w}, \boldsymbol{v}(\boldsymbol{w}, t), t) = \mathbf{0}$,
$$\nabla_{\boldsymbol{w}}u(\boldsymbol{w}, t) = \nabla_{\boldsymbol{w}}z(\boldsymbol{w}, \boldsymbol{v}(\boldsymbol{w}, t), t) = -\frac{1}{t}\boldsymbol{A}_\sigma(\boldsymbol{v}(\boldsymbol{w}, t) - \boldsymbol{w}).$$
Letting $\boldsymbol{w} = \boldsymbol{w}^k$ and $\boldsymbol{w}^{k+1} = \boldsymbol{v}(\boldsymbol{w}^k, t) = \arg\min_{\boldsymbol{v}} z(\boldsymbol{w}^k, \boldsymbol{v}, t)$ in the above equalities, we have
$$\nabla_{\boldsymbol{w}}u(\boldsymbol{w}^k, t) = -\frac{1}{t}\boldsymbol{A}_\sigma(\boldsymbol{w}^{k+1} - \boldsymbol{w}^k).$$
In summary, the gradient descent $\boldsymbol{w}^{k+1} = \boldsymbol{w}^k - t\boldsymbol{A}_\sigma^{-1}\nabla_{\boldsymbol{w}}u(\boldsymbol{w}^k, t)$ is equivalent to the proximal point iteration $\boldsymbol{w}^{k+1} = \arg\min_{\boldsymbol{v}} f(\boldsymbol{v}) + \frac{1}{2t}\langle \boldsymbol{v} - \boldsymbol{w}^k, \boldsymbol{A}_\sigma(\boldsymbol{v} - \boldsymbol{w}^k)\rangle$, which yields $\boldsymbol{w}^{k+1} = \boldsymbol{w}^k - t\boldsymbol{A}_\sigma^{-1}\nabla f(\boldsymbol{w}^{k+1})$. □

**Proposition 2.** *Suppose $f$ is convex with the global minimizer $\boldsymbol{w}^*$, and $f^* = f(\boldsymbol{w}^*)$. Consider the following iteration with constant learning rate $\eta > 0$*
$$\boldsymbol{w}^{k+1} = \boldsymbol{w}^k - \eta(\boldsymbol{A}_\sigma^n)^{-1}\boldsymbol{g}^k$$
*where $\boldsymbol{g}^k$ is the sampled gradient in the $k$th iteration at $\boldsymbol{w}^k$ satisfying $\mathbb{E}[\boldsymbol{g}^k] = \nabla f(\boldsymbol{w}^k)$. Denote $G_{\boldsymbol{A}_\sigma^n} := \lim_{K\to\infty} \frac{1}{K}\sum_{k=0}^{K-1}\|\boldsymbol{g}^k\|_{(\boldsymbol{A}_\sigma^n)^{-1}}^2$ and $\overline{\boldsymbol{w}}^K := \sum_{k=0}^{K-1}\boldsymbol{w}^k/K$ the ergodic average of iterates. Then the optimality gap is*
$$\lim_{K\to\infty}\mathbb{E}[f(\overline{\boldsymbol{w}}^K)] - f^* \leq \frac{\eta G_{\boldsymbol{A}_\sigma^n}}{2}.$$

*Proof.* Since $f$ is convex, we have
$$\langle\nabla f(\boldsymbol{w}^k), \boldsymbol{w}^k - \boldsymbol{w}^*\rangle \geq f(\boldsymbol{w}^k) - f^*. \tag{7}$$
Furthermore,
$$\mathbb{E}[\|\boldsymbol{w}^{k+1} - \boldsymbol{w}^*\|_{\boldsymbol{A}_\sigma^n}^2] = \mathbb{E}[\|\boldsymbol{w}^k - \eta(\boldsymbol{A}_\sigma^n)^{-1}\boldsymbol{g}^k - \boldsymbol{w}^*\|_{\boldsymbol{A}_\sigma^n}^2]$$
$$= \mathbb{E}[\|\boldsymbol{w}^k - \boldsymbol{w}^*\|_{\boldsymbol{A}_\sigma^n}^2] - 2\eta\mathbb{E}[\langle\boldsymbol{g}^k, \boldsymbol{w}^k - \boldsymbol{w}^*\rangle] + \eta^2\mathbb{E}[\|(\boldsymbol{A}_\sigma^n)^{-1}\boldsymbol{g}^t\|_{\boldsymbol{A}_\sigma^n}^2]$$
$$\leq \mathbb{E}[\|\boldsymbol{w}^k - \boldsymbol{w}^*\|_{\boldsymbol{A}_\sigma^n}^2] - 2\eta\mathbb{E}[\langle\nabla f(\boldsymbol{w}^k), \boldsymbol{w}^k - \boldsymbol{w}^*\rangle] + \eta^2\|\boldsymbol{g}^k\|_{(\boldsymbol{A}_\sigma^n)^{-1}}^2$$
$$\leq \mathbb{E}[\|\boldsymbol{w}^k - \boldsymbol{w}^*\|_{\boldsymbol{A}_\sigma^n}^2] - 2\eta(\mathbb{E}[f(\boldsymbol{w}^k)] - f^*) + \eta^2\|\boldsymbol{g}^k\|_{(\boldsymbol{A}_\sigma^n)^{-1}}^2,$$
where the last inequality is due to (7). We rearrange the terms and arrive at
$$\mathbb{E}[f(\boldsymbol{w}^k)] - f^* \leq \frac{1}{2\eta}(\mathbb{E}[\|\boldsymbol{w}^k - \boldsymbol{w}^*\|_{\boldsymbol{A}_\sigma^n}^2] - \mathbb{E}[\|\boldsymbol{w}^{k+1} - \boldsymbol{w}^*\|_{\boldsymbol{A}_\sigma^n}^2]) + \frac{\eta\|\boldsymbol{g}^k\|_{(\boldsymbol{A}_\sigma^n)^{-1}}^2}{2}.$$
Summing over $k$ from 0 to $K-1$ and averaging and using the convexity of $f$, we have
$$\mathbb{E}[f(\overline{\boldsymbol{w}}^K)] - f^* \leq \frac{\sum_{k=0}^{K-1}\mathbb{E}[f(\boldsymbol{w}^k)]}{K} - f^* \leq \frac{1}{2\eta K}\mathbb{E}[\|\boldsymbol{w}^0 - \boldsymbol{w}^*\|_{\boldsymbol{A}_\sigma^n}^2] + \frac{\sum_{k=0}^{K-1}\|\boldsymbol{g}^k\|_{(\boldsymbol{A}_\sigma^n)^{-1}}^2}{2K}\eta.$$
Taking the limit as $K \to \infty$ above establishes the result. □

**Proposition 3.** *Suppose $f$ is $L$-Lipschitz smooth and $a$-strongly convex. Consider the generalized smoothing gradient descent algorithm*

$$\boldsymbol{w}^{k+1} = \boldsymbol{w}^k - \eta_k (\boldsymbol{A}_\sigma^n)^{-1} \boldsymbol{g}^k,$$

*where $\boldsymbol{g}^k$ is the sampled gradient in the $k$th iteration at $\boldsymbol{w}^k$ satisfying $\mathbb{E}\left[\boldsymbol{g}^k\right] = \nabla f(\boldsymbol{w}^k)$ and $\mathbb{E}\left[\|\boldsymbol{g}^k\|^2_{(\boldsymbol{A}_\sigma^n)^{-1}}\right] \leq C_0 + C_1 \|\nabla f(\boldsymbol{w}^k)\|^2$ for all $k \in \mathbb{N}$. If we take $\eta_k = \frac{C}{k+1}$ for some $C > 0$, then we have*

$$\mathbb{E}\left[\|\boldsymbol{w}^k - \boldsymbol{w}^*\|^2_{\boldsymbol{A}_\sigma^n}\right] = \mathbb{E}\left[\|\boldsymbol{w}^k - \boldsymbol{w}^*\|^2 + \sigma \|\boldsymbol{D}_+^n(\boldsymbol{w}^k - \boldsymbol{w}^*)\|^2\right] = O\left(\frac{1}{k+1}\right),$$

*i.e., we have $H_\sigma^n$ uniform convergence in $\sigma$ of $\{\boldsymbol{w}^k\}$ in expectation. The $H_\sigma^n$ norm of $\boldsymbol{w}$ is defined by $\|\boldsymbol{w}\|_\sigma^n := \|w\|_{\boldsymbol{A}_\sigma^n} = \sqrt{\langle \boldsymbol{w}, \boldsymbol{A}_\sigma^n \boldsymbol{w}\rangle}$.*

**Proof of Proposition 3.** Since $\nabla f(\boldsymbol{w}^*) = \boldsymbol{0}$, by strong convexity of $f$, we have

$$\langle \nabla f(\boldsymbol{w}^k), \boldsymbol{w}^k - \boldsymbol{w}^*\rangle = \langle \nabla f(\boldsymbol{w}^k) - \nabla f(\boldsymbol{w}^*), \boldsymbol{w}^k - \boldsymbol{w}^*\rangle \geq a\|\boldsymbol{w}^k - \boldsymbol{w}^*\|^2.$$

Moreover, by $L$-smoothness of $f$ and the fact that $\|\boldsymbol{A}_\sigma^n\| = 1$, we also have

$$\|\nabla f(\boldsymbol{w}^k)\| = \|\nabla f(\boldsymbol{w}^k) - \nabla f(\boldsymbol{w}^*)\| \leq L\|\boldsymbol{w}^k - \boldsymbol{w}^*\| \leq L\|\boldsymbol{w}^k - \boldsymbol{w}^*\|_{\boldsymbol{A}_\sigma^n}.$$

Hence,

$$\begin{aligned}
\mathbb{E}[\|\boldsymbol{w}^{k+1} - \boldsymbol{w}^*\|^2_{\boldsymbol{A}_\sigma^n}] &= \mathbb{E}[\|\boldsymbol{w}^k - \eta(\boldsymbol{A}_\sigma^n)^{-1}\boldsymbol{g}^k - \boldsymbol{w}^*\|^2_{\boldsymbol{A}_\sigma^n}] \\
&= \mathbb{E}[\|\boldsymbol{w}^k - \boldsymbol{w}^*\|^2_{\boldsymbol{A}_\sigma^n}] - 2\eta_k \mathbb{E}\left[\langle \boldsymbol{g}^k, \boldsymbol{w}^k - \boldsymbol{w}^*\rangle\right] + \eta_k^2 \mathbb{E}[\|\boldsymbol{g}^k\|^2_{(\boldsymbol{A}_\sigma^n)^{-1}}] \\
&= \mathbb{E}[\|\boldsymbol{w}^k - \boldsymbol{w}^*\|^2_{\boldsymbol{A}_\sigma^n}] - 2\eta_k \langle \nabla f(\boldsymbol{w}^k), \boldsymbol{w}^k - \boldsymbol{w}^*\rangle + \eta_k^2 \mathbb{E}[\|\boldsymbol{g}^k\|^2_{(\boldsymbol{A}_\sigma^n)^{-1}}] \\
&\leq (1 - 2\eta_k a)\mathbb{E}\left[\|\boldsymbol{w}^k - \boldsymbol{w}^*\|^2_{\boldsymbol{A}_\sigma^n}\right] + \eta_k^2 \left(C_0 + C_1 \mathbb{E}[\|\nabla f(\boldsymbol{w}^k)\|^2]\right) \\
&\leq \left(1 - 2\eta_k a + \eta_k^2 L^2 C_1\right) \mathbb{E}\left[\|\boldsymbol{w}^k - \boldsymbol{w}^*\|^2_{\boldsymbol{A}_\sigma^n}\right] + \eta_k^2 C_0,
\end{aligned}$$

where in the first inequality we used $\|(\boldsymbol{A}_\sigma^n)^{-1}\| = 1$ for all $\sigma$ and $n$. Taking $\eta_k = \frac{C}{k+1}$ for some proper $C > 0$ and using induction, one can show that $\mathbb{E}\left[\|\boldsymbol{w}^k - \boldsymbol{w}^*\|^2_{\boldsymbol{A}_\sigma^n}\right] = \mathbb{E}\left[\|\boldsymbol{w}^k - \boldsymbol{w}^*\|^2 + \sigma\|\boldsymbol{D}_+^n(\boldsymbol{w}^k - \boldsymbol{w}^*)\|\right] = O(\frac{1}{k+1})$. $\qquad\square$

**Proposition 4.** *Consider the algorithm $\boldsymbol{w}^{k+1} = \boldsymbol{w}^k - \eta_k (\boldsymbol{A}_\sigma^n)^{-1}\nabla f(\boldsymbol{w}^k)$. Suppose $f$ is $L$-Lipschitz smooth and $0 < \underline{\eta} \leq \eta \leq \bar{\eta} < \frac{2}{L}$. Then $\lim_{t\to\infty}\|\nabla f(\boldsymbol{w}^k)\| \to 0$. Moreover, if the Hessian $\nabla^2 f$ of $f$ is continuous with $\boldsymbol{w}^*$ being the minimizer of $f$, and $\bar{\eta}\|\nabla^2 f\| < 1$, then $\|\boldsymbol{w}^k - \boldsymbol{w}^*\|_{\boldsymbol{A}_\sigma^n} \to 0$ as $k \to \infty$, and the convergence is linear.*

**Proof of Proposition 4.** By the Lipschitz continuity of $\nabla f$ and the descent lemma (Bertsekas, 1999), we have

$$\begin{aligned}
f(\boldsymbol{w}^{k+1}) &= f(\boldsymbol{w}^k - \eta_k (\boldsymbol{A}_\sigma^n)^{-1}\nabla f(\boldsymbol{w}^k)) \\
&\leq f(\boldsymbol{w}^k) - \eta_k \langle \nabla f(\boldsymbol{w}^k), (\boldsymbol{A}_\sigma^n)^{-1}\nabla f(\boldsymbol{w}^k)\rangle + \frac{\eta_k^2 L}{2}\|(\boldsymbol{A}_\sigma^n)^{-1}\nabla f(\boldsymbol{w}^k)\|^2 \\
&\leq f(\boldsymbol{w}^k) - \eta_k \|\nabla f(\boldsymbol{w}^k)\|^2_{(\boldsymbol{A}_\sigma^n)^{-1}} + \frac{\eta_k^2 L}{2}\|\nabla f(\boldsymbol{w}^k)\|^2_{(\boldsymbol{A}_\sigma^n)^{-1}} \\
&\leq f(\boldsymbol{w}^k) - \underline{\eta}\left(1 - \frac{\bar{\eta}L}{2}\right)\|\nabla f(\boldsymbol{w}^k)\|^2_{(\boldsymbol{A}_\sigma^n)^{-1}}.
\end{aligned}$$

Summing the above inequality over $k$, we have

$$\underline{\eta}\left(1 - \frac{\bar{\eta}L}{2}\right)\sum_{k=0}^\infty \|\nabla f(\boldsymbol{w}^k)\|^2_{(\boldsymbol{A}_\sigma^n)^{-1}} \leq f(\boldsymbol{w}^0) - \lim_{k\to\infty} f(\boldsymbol{w}^k) < \infty.$$

Therefore, $\|\nabla f(\boldsymbol{w}^k)\|^2_{(\boldsymbol{A}^n_\sigma)^{-1}} \to 0$, and thus $\|\nabla f(\boldsymbol{w}^k)\| \to 0$.

For the second claim, we have

$$\boldsymbol{w}^{k+1} - \boldsymbol{w}^* = \boldsymbol{w}^k - \boldsymbol{w}^* - \eta_k(\boldsymbol{A}^n_\sigma)^{-1}(\nabla f(\boldsymbol{w}^k) - \nabla f(\boldsymbol{w}^*))$$

$$= \boldsymbol{w}^k - \boldsymbol{w}^* - \eta_k(\boldsymbol{A}^n_\sigma)^{-1}\left(\int_0^1 \nabla^2 f(\boldsymbol{w}^* + \tau(\boldsymbol{w}^{k+1} - \boldsymbol{w}^*)) \cdot (\boldsymbol{w}^k - \boldsymbol{w}^*)\mathrm{d}\tau\right)$$

$$= \boldsymbol{w}^k - \boldsymbol{w}^* - \eta_k(\boldsymbol{A}^n_\sigma)^{-1}\left(\int_0^1 \nabla^2 f(\boldsymbol{w}^* + \tau(\boldsymbol{w}^{k+1} - \boldsymbol{w}^*))\mathrm{d}\tau \cdot (\boldsymbol{w}^k - \boldsymbol{w}^*)\right)$$

$$= (\boldsymbol{A}^n_\sigma)^{-\frac{1}{2}}\left(\boldsymbol{I} - \eta_k(\boldsymbol{A}^n_\sigma)^{-\frac{1}{2}}\int_0^1 \nabla^2 f(\boldsymbol{w}^* + \tau(\boldsymbol{w}^{k+1} - \boldsymbol{w}^*))\mathrm{d}\tau(\boldsymbol{A}^n_\sigma)^{-\frac{1}{2}}\right)(\boldsymbol{A}^n_\sigma)^{\frac{1}{2}}(\boldsymbol{w}^k - \boldsymbol{w}^*)$$

Therefore,

$$\|\boldsymbol{w}^{k+1} - \boldsymbol{w}^*\|_{\boldsymbol{A}^n_\sigma} \leq \left\|\boldsymbol{I} - \eta_t(\boldsymbol{A}^n_\sigma)^{-\frac{1}{2}}\int_0^1 \nabla^2 f(\boldsymbol{w}^* + \tau(\boldsymbol{w}^{k+1} - \boldsymbol{w}^*))\mathrm{d}\tau(\boldsymbol{A}^n_\sigma)^{-\frac{1}{2}}\right\|\|\boldsymbol{w}^k - \boldsymbol{w}^*\|_{\boldsymbol{A}^n_\sigma}.$$

So if $\eta_k\|\nabla^2 f\| \leq \frac{1}{\|(\boldsymbol{A}^n_\sigma)^{-1}\|} = 1$, the result follows. $\qquad\square$

**Proposition 5.** *For any vector* $\boldsymbol{g} \in \mathbb{R}^m$, $\boldsymbol{d} = \boldsymbol{A}^{-1}_\sigma\boldsymbol{g}$, *let* $j_{\max} = \arg\max_i d_i$ *and* $j_{\min} = \arg\min_i d_i$. *We have* $\max_i d_i = d_{j_{\max}} \leq g_{j_{\max}} \leq \max_i g_i$ *and* $\min_i d_i = d_{j_{\min}} \geq g_{j_{\min}} \geq \min_i g_i$.

**Proof of Proposition 5.** Since $\boldsymbol{g} = \boldsymbol{A}_\sigma\boldsymbol{d}$, it holds that

$$g_{j_{\max}} = d_{j_{\max}} + \sigma(2d_{j_{\max}} - d_{j_{\max}-1} - d_{j_{\max}+1}),$$

where periodicity of subindex are used if necessary. Since $2d_{j_{\max}} - d_{j_{\max}-1} - d_{j_{\max}+1} \geq 0$, We have $\max_i d_i = d_{j_{\max}} \leq g_{j_{\max}} \leq \max_i g_i$. A similar argument can show that $\min_i d_i = d_{j_{\min}} \geq g_{j_{\min}} \geq \min_i g_i$. $\qquad\square$

**Proposition 6.** *The operator* $\boldsymbol{A}^{-1}_\sigma$ *preserves the sum of components. For any* $\boldsymbol{g} \in \mathbb{R}^m$ *and* $\boldsymbol{d} = \boldsymbol{A}^{-1}_\sigma\boldsymbol{g}$, *we have* $\sum_j d_j = \sum_j g_j$, *or equivalently,* $\mathbf{1}^\top\boldsymbol{d} = \mathbf{1}^\top\boldsymbol{g}$.

**Proof of Proposition 6.** Since $\boldsymbol{g} = \boldsymbol{A}_\sigma\boldsymbol{d}$,

$$\sum_i g_i = \mathbf{1}^\top\boldsymbol{g} = \mathbf{1}^\top(\boldsymbol{I} + \sigma\boldsymbol{D}^\top_+\boldsymbol{D}_+)\boldsymbol{d} = \mathbf{1}^\top\boldsymbol{d} = \sum_i d_i,$$

where we used $\boldsymbol{D}_+\mathbf{1} = \mathbf{0}$. $\qquad\square$

**Proposition 7.** *Given any vector* $\boldsymbol{g} \in \mathbb{R}^m$ *and* $\boldsymbol{d} = \boldsymbol{A}^{-1}_\sigma\boldsymbol{g}$, *then*

$$\|\boldsymbol{d}\| + \sigma\frac{\|\boldsymbol{D}_+\boldsymbol{d}\|^2}{\|\boldsymbol{d}\|} \leq \|\boldsymbol{g}\|.$$

*The above inequality is strict unless* $\boldsymbol{g} = \boldsymbol{d}$ *is a constant vector. In particular, we have* $\|\boldsymbol{d}\| \leq \|\boldsymbol{g}\|$ *and* $\|\boldsymbol{D}_+\boldsymbol{d}\| \leq \frac{1}{\sqrt{\sigma}}\|\boldsymbol{g}\|$.

**Proof of Proposition 7.** By the definition of $\boldsymbol{A}_\sigma$,

$$\boldsymbol{g} = \boldsymbol{A}_\sigma\boldsymbol{d} = (\boldsymbol{I} - \sigma\boldsymbol{D}_-\boldsymbol{D}_+)\boldsymbol{d} = \boldsymbol{d} + \sigma\boldsymbol{D}^\top_+\boldsymbol{D}_+\boldsymbol{d}. \tag{8}$$

Therefore, pre-multiplying by $\boldsymbol{d}^\top$ on both sides, we have

$$\|\boldsymbol{d}\|^2 + \sigma\|\boldsymbol{D}_+\boldsymbol{d}\|^2 = \boldsymbol{d}^\top\boldsymbol{g} \leq \|\boldsymbol{d}\|\|\boldsymbol{g}\|.$$

In particular, $\|\boldsymbol{d}\| \leq \|\boldsymbol{g}\|$ and $\sigma\|\boldsymbol{D}_+\boldsymbol{d}\|^2 \leq \|\boldsymbol{d}\|\|\boldsymbol{g}\| \leq \|\boldsymbol{g}\|^2$, so $\|\boldsymbol{D}_+\boldsymbol{d}\| \leq \frac{1}{\sqrt{\sigma}}\|\boldsymbol{g}\|$. All the inequalities are strict unless $\|\boldsymbol{D}_+\boldsymbol{d}\| = 0$, and $\boldsymbol{g} = \boldsymbol{d}$ is a constant vector. $\qquad\square$

**Proposition 8.** *For any $\boldsymbol{g} \in \mathbb{R}^m$, define $\mathrm{Var}(\boldsymbol{g}) := \frac{1}{m}\|\boldsymbol{g}\|^2 - \left(\frac{\mathbf{1}^\top \boldsymbol{g}}{m}\right)^2$ be the variance of components in $\boldsymbol{g}$. Let $\boldsymbol{d} = \boldsymbol{A}_\sigma^{-1}\boldsymbol{g}$, then*

$$\mathrm{Var}(\boldsymbol{d}) \leq \mathrm{Var}(\boldsymbol{g}) - 2\sigma\frac{\|\boldsymbol{D}_+\boldsymbol{d}\|^2}{m} - \sigma^2\frac{\|\boldsymbol{D}_+\boldsymbol{d}\|^4}{m\|\boldsymbol{d}\|^2}.$$

*The inequality is strict unless $\boldsymbol{g} = \boldsymbol{d}$ is a constant vector.*

**Proof of Proposition 8.** Since $\mathbf{1}^\top \boldsymbol{g} = \mathbf{1}^\top \boldsymbol{d}$ and $\|\boldsymbol{d}\| + \sigma\frac{\|\boldsymbol{D}_+\boldsymbol{d}\|^2}{\|\boldsymbol{d}\|} \leq \|\boldsymbol{g}\|$,

$$\mathrm{Var}(\boldsymbol{g}) \geq \frac{1}{m}\left(\|\boldsymbol{d}\|^2 + 2\sigma\|\boldsymbol{D}_+\boldsymbol{d}\|^2 + \sigma^2\frac{\|\boldsymbol{D}_+\boldsymbol{d}\|^4}{\|\boldsymbol{d}\|^2}\right) - \left(\frac{\mathbf{1}^\top \boldsymbol{d}}{n}\right)^2$$

$$= \mathrm{Var}(\boldsymbol{d}) + 2\sigma\frac{\|\boldsymbol{D}_+\boldsymbol{d}\|^2}{m} + \sigma^2\frac{\|\boldsymbol{D}_+\boldsymbol{d}\|^4}{m\|\boldsymbol{d}\|^2}.$$

The inequality is strict unless $\|\boldsymbol{D}_+\boldsymbol{d}\| = 0$, and $\boldsymbol{g} = \boldsymbol{d}$ is a constant vector. $\qquad\square$

**Proposition 9.** *Given vectors $\boldsymbol{g}$ and $\boldsymbol{d} = \boldsymbol{A}_\sigma^{-1}\boldsymbol{g}$, for any $p \in \mathbb{N}$, it holds that $\|\boldsymbol{D}_+^p\boldsymbol{d}\|_1 \leq \|\boldsymbol{D}_+^p\boldsymbol{g}\|_1$. The inequality is strict unless $\boldsymbol{D}_+^p\boldsymbol{g}$ is a constant vector.*

**Proof of Proposition 9.** Since $(1 + 2\sigma)d_i = g_i + \sigma d_{i+1} + \sigma d_{i-1}$, for any $p \in \mathbb{N}$, we have

$$(1 + 2\sigma)(\boldsymbol{D}_+^p\boldsymbol{d})_i = (\boldsymbol{D}_+^p\boldsymbol{g})_i + \sigma(\boldsymbol{D}_+^p\boldsymbol{d})_{i+1} + \sigma(\boldsymbol{D}_+^p\boldsymbol{d})_{i-1}.$$

So

$$(1 + 2\sigma)|(\boldsymbol{D}_+^p\boldsymbol{d})_i| \leq |(\boldsymbol{D}_+^p\boldsymbol{g})_i| + \sigma|(\boldsymbol{D}_+^p\boldsymbol{d})_{i+1}| + \sigma|(\boldsymbol{D}_+^p\boldsymbol{d})_{i-1}|.$$

The inequality is strict if there are sign changes among the $(\boldsymbol{D}_+^p\boldsymbol{d})_{i-1}$, $(\boldsymbol{D}_+^p\boldsymbol{d})_i$, $(\boldsymbol{D}_+^p\boldsymbol{d})_{i+1}$. Summing over $i$ and using periodicity, we have

$$(1 + 2\sigma)\sum_{i=1}^m |(\boldsymbol{D}_+^p\boldsymbol{d})_i| \leq \sum_{i=1}^m |(\boldsymbol{D}_+^p\boldsymbol{g})_i| + 2\sigma\sum_{i=1}^m |(\boldsymbol{D}_+^p\boldsymbol{d})_i|,$$

and the result follows. The inequality is strict unless $\boldsymbol{D}_+^p\boldsymbol{g}$ is a constant vector.

$\square$

## 6.2 Iteration v.s. Loss for Softmax Regression

In this part, we show the training loss evolution in training Softmax regression model, respectively, by SGD, SVRG, LSGD with first and second order smoothing. As illustrated in Fig. 10, all the optimization algorithms reduce loss of the model on the training set. At each iteration, among 100 independent experiments, SGD has the largest variance, SGD with first order smoothed gradient significantly reduces the variance of loss function. The second order smoothing can further reduce variance of loss. The variance of loss in each iteration among 100 experiments is minimized when SVRG is use to train the Softmax model. However, the generalization performance of the model trained by SVRG is not as good as the ones trained by LS-SGD or higher order smoothed gradient descent.

## 6.3 Deep Reinforcement Learning

Deep reinforcement learning (DRL) has been applied to playing games including Cartpole (Brockman et al., 2016), Atari (Mnih et al., 2013), Go (Silver & et al, 2016; Mnih & et al, 2015). DNN plays a vital role in approximating the Q-function or policy function. We apply the Laplacian smoothed gradient to train the policy function to play the Cartpole game. We apply the standard procedure to train the policy function by using the policy gradient (Brockman et al., 2016). We use the following network to approximate the policy function:

$$\mathrm{input}_4 \to \mathrm{fc}_{20} \to \mathrm{relu} \to \mathrm{fc}_2 \to \mathrm{softmax}.$$

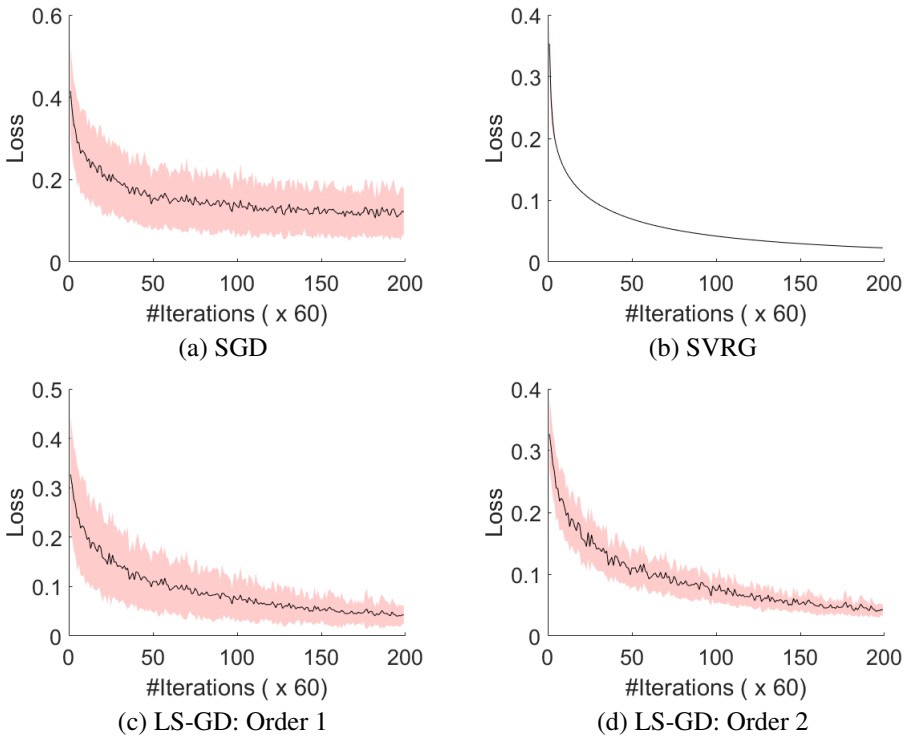

(a) SGD  (b) SVRG

(c) LS-GD: Order 1  (d) LS-GD: Order 2

Figure 10: Iterations v.s. loss for GD, SVRG, and smoothed GD with order 1 and 2 for training the softmax regression model.

The network is trained by RMSProp and LS-RMSProp with $\sigma = 1.0$, respectively. The learning rate and other related parameters are set to be the default ones in PyTorch. The training is stopped once the average duration of 5 consecutive episodes is more than 490. In each training episode, we set the maximal steps to be 500. Left and right panels of Fig. 11 depict a training procedure by using RMSProp and LS-RMSProp, respectively. We see that Laplacian smoothed gradient takes fewer episodes to reach the stopping criterion. Moreover, we run the above experiments 5 times independently, and apply the trained model to play Cartpole. The game lasts more than 1000 steps for all the 5 models trained by LS-RMSProp, while only 3 of them lasts more than 1000 steps when the model is trained by vanilla RMSProp.

RMSProp  LS-RMSProp, $\sigma = 1.0$

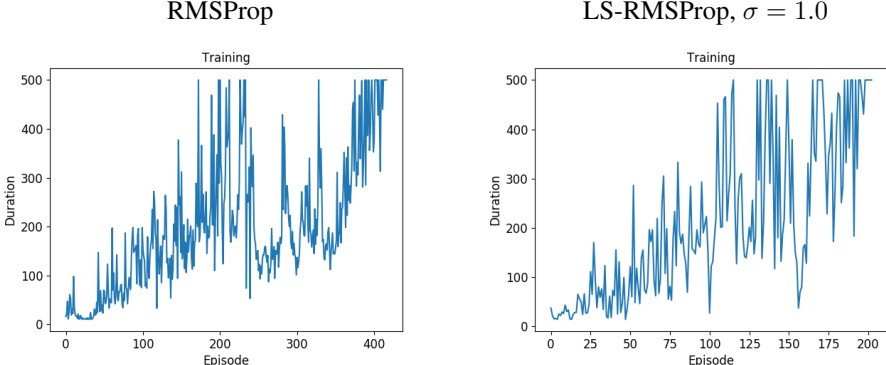

Figure 11: Durations of the cartpole game in the training procedure. Left and right are training procedure by RMSProp and LS-RMSProp with $\sigma = 1.0$, respectively.

