# OpenReview forum: "Laplacian Smoothing Gradient Descent"
_ICLR.cc/2019/Conference_

### Official Review · AnonReviewer1 · 2018-10-31
**Some concerns on experiments and written style**

**Rating:** 5
**Confidence:** 4

**Review:**

The paper proposes to use a simple tri-diagonal matrix to reduce the variance of stochastic gradient and provide a better generalization property. Such a variant is shown to be equivalent to applying GD on smoothed objective function. Theoretical results show a convergence rate and variance reduction. Various experiments are done in different settings.  I have following comments:

1) In section 2, it is stated that "This viscosity solution u(w, t) makes f(w) more convex by bringing down the local maxima while retaining the wide minima." Besides illustrating such a point on some nicely constructed function f, is there any theory or analysis supporting this statement? Or is there any intuition behind it? In the abstract and Section 1, how to define a function is "more convex"? This is one of the fountains of the paper, it worths to spend one or two paragraphs to explain it, or at least introduce some references here. The current statement is not formed in a rigorous way.

2) The main advantages of proposed method that the paper claims are, reduce the variance and improve the generalization accuracy. However, there are few comparisons with other existed methods, besides numerical section. Such comparisons or analysis could help readers understand the difference and novelty.

3) The proof seems fine. Propositions 1-4 try to analyze the convergence rate, which are common techniques in other variance reduction papers on SGD. Propositions 5-9 rely on some nice properties of matrix A_\sigma and show it can help to reduce the variance. Typos:
Page 11, "Proof of Proposition 1", there is a missing "-" in \nabla_w u(w, t), also in the next equation.
Page 13, "Proof of Proposition 6, d = A_\sigma g".

4) The proposed method strongly relies on the choice of \sigma, but discussion on how to choose the value for \sigma is rare. From Proposition 8, the upper bound on reduced variance is a quadratic function on \sigma, so it is better to discuss more on it or have some experiments on sensitivity analysis. In Section 4, \sigma varies (1.0, 3.0, etc) in different experiments, but again there are no explanations.

5) Numerical results in Section 4.3 is not strong enough to support the advantage of the proposed method. It is hard to observe "visibly less noisy" in both Figure 8 and 9. Better ways of illustration might be considered.

6) The paper is not nicely written thus cannot be easily read. It seems to be cut and pasted from another version in a short time. Some titles of subsections is missing. The font size is not fixed in the whole paper.

The above concerns prevents me to give a higher rating at this time.

Summary
quality: ok
clarity: good
originality: nice
significance: good

---

> ### Author Response · Authors · 2018-11-17
> **Response to Reviewer #1**
>
> Thanks for the detailed and thoughtful review. We've responded to your comments below. Based on your comments we have modified our paper from several aspects. As you will see, your review helps us to improve the manuscript significantly.  We would be very grateful if you would look over our paper again, and reconsider your opinion.
>
> Q1. In section 2, it is stated that "This viscosity solution u(w, t) makes f(w) more convex by bringing down the local maxima while retaining the wide minima." Besides illustrating such a point on some nicely constructed function f, is there any theory or analysis supporting this statement? Or is there any intuition behind it? In the abstract and Section 1, how to define a function is "more convex"? This is one of the fountains of the paper, it worths to spend one or two paragraphs to explain it, or at least introduce some references here. The current statement is not formed in a rigorous way.
>
> A: Thanks for your helpful suggestion. The intuitive definition and theoretical explanation of making the loss function “more convex” can be found in the papers: Chaudhari, et al., ICLR, 2016 and Chaudhari, et al., arXiv:1704.04932, 2017. In the revised paper, we have clarified this in Section 2.
>
>
> Q2. The main advantages of the proposed method that the paper claims are, reduce the variance and improve the generalization accuracy. However, there are few comparisons with other existed methods, besides numerical section. Such comparisons or analysis could help readers understand the difference and novelty.
>
> A: To our knowledge, the existing variance reduction algorithms mostly depend on either the reference gradient or the stochastic gradient for a large amount of data. We achieve this by some simple manipulation of the stochastic gradient, which has very little extra overhead compared to the standard SGD. Combining Laplacian smoothing with existing variance reduction to improve performance and theoretical bounds are being explored by us. The theory of generalization for general deep neural nets is still unclear, but to justify our proposed algorithm, we performed a large amount of testing on challenging deep learning-related tasks. This experimental justification received appreciation from some reviewers.
>
>
> Q3. The proof seems fine. Propositions 1-4 try to analyze the convergence rate, which are common techniques in other variance reduction papers on SGD. Propositions 5-9 rely on some nice properties of matrix A_\sigma and show it can help to reduce the variance. Typos: Page 11, "Proof of Proposition 1", there is a missing "-" in \nabla_w u(w, t), also in the next equation. Page 13, "Proof of Proposition 6, d = A_\sigma g".
>
> A: Thanks for pointing out these typos. We have fixed them in the updated manuscript.
>
>
> Q4. The proposed method strongly relies on the choice of \sigma, but discussion on how to choose the value for \sigma is rare. From Proposition 8, the upper bound on reduced variance is a quadratic function on \sigma, so it is better to discuss more on it or have some experiments on sensitivity analysis. In Section 4, \sigma varies (1.0, 3.0, etc) in different experiments, but again there are no explanations.
>
> A: In our experiments, we typically set sigma = 1.0, which always show some improvement over the (stochastic) gradient descent. For the Wasserstein GAN experiment, we tried a few different sigmas, and all have a certain amount of improvement over the original RMSProp optimizer. In general, big sigma removes more noise from the stochastic gradient. But in training neural nets, we often need a certain amount of noise. Usually, we default sigma to be 1.
>
>
> Q5. Numerical results in Section 4.3 is not strong enough to support the advantage of the proposed method. It is hard to observe "visibly less noisy" in both Figure 8 and 9. Better ways of illustration might be considered.
>
> A: Thanks for the suggestion. Figure. 8 shows that the training loss curve becomes less spiky when the Laplacian smoothing is applied to the stochastic gradient. For Fig. 9, if we look at the images in panel (a) and (b), we see that images in panel (b) are more realistic. In Fig.9(a), there are more unrealistic digits generated by the neural nets compared to the panel (b). We have stressed this in the revised manuscript.
>
>
> Q6. The paper is not nicely written thus cannot be easily read. It seems to be cut and pasted from another version in a short time. Some titles of subsections are missing. The font size is not fixed in the whole paper.
>
> A: We are sorry to let you feel our paper is hard to read. In the revised version, we have improved the writing and fixed the font size.

---

### Official Review · AnonReviewer2 · 2018-11-02
**Theory and experiments can be improved**

**Rating:** 6
**Confidence:** 4

**Review:**

This paper proposed a variant of gradient descent that can be approximately understood as gradient descent on a smoothed version of the objective function. The motivation of this work is finding flat minima which could imply better generalization ability of a machine learning model.
Compared with gradient, the proposed algorithm uses gradient multiplied by a special constant square matrix as update direction. The complexity of the matrix vector multiplication is brought down from O(d^2) to O(d*logd) by exploiting special structure of the matrix using FFT. It is proved that the new update vector has smaller variance and amplitude compared to gradient.
Experiments on different applications showed that the proposed algorithm may have better generalization ability compared with SGD.
This is a clearly written paper, but I have a few questions about theoretical gaps and simulation results in the paper.

a). It seems the smoothing explanation at the beginning of section 2 is for implicit scheme (equation (3)). However, the explicit scheme used in practice (the first unnumbered equation in section 2.1) uses a heuristic relaxation which makes the smoothing explanation “approximate” for the explicit scheme. Since the implicit scheme is much more complicated than the explicit scheme, I don’t know if the argument for the implicit scheme will “approximately” hold for the explicit scheme used in practice.

b). The concept flat minimum is only useful in nonconvex optimization, but the convergence of the algorithm is only proved in convex setting. Since the main motivation of the algorithm is finding flat minima, the lack of convergence proof for nonconvex setting concerns me.

c). In the neural net experiment in section 4.1, both gradient descent and smooth gradient descent use the same stepsizes. It is known that the performance of gradient descent is sensitive to the choice of stepsizes, for a fair comparison, one should compare the performance of the two algorithms using optimized stepsizes.

d). In the experiment in section 4.2, the proposed algorithm is only used for the first 40 epochs during training and SGD is used for the later phase of training. Why switching to SGD later?

Overall, I feel the idea of this paper is interesting, but the theory and experiments in the paper are not very strong.

---

> ### Author Response · Authors · 2018-11-17
> **Response to Reviewer #2**
>
> We are thankful for the detailed reading and careful evaluation of our work. We have revised the manuscript to address the reviewer’s comments.
>
> Q1. It seems the smoothing explanation at the beginning of section 2 is for an implicit scheme (equation (3)). However, the explicit scheme used in practice (the first unnumbered equation in section 2.1) uses a heuristic relaxation which makes the smoothing explanation “approximate” for the explicit scheme. Since the implicit scheme is much more complicated than the explicit scheme, I don’t know if the argument for the implicit scheme will “approximately” hold for the explicit scheme used in practice.
>
> A: Thanks for pointing out this. The Moreau envelope only serves as a motivation of our algorithm. Another motivation of our algorithm is to reduce the variance of stochastic gradient on-the-fly. All of our arguments on variance reduction and smoothing properties do not depend on the implicit scheme. For training deep neural nets part, we verify the efficacy of the proposed LS-SGD by a large number of experiments. Moreover, the result in Figure 2 shows that Laplacian smoothing helps to circumvent local spurious minima.
>
>
> Q2. The concept flat minimum is only useful in nonconvex optimization, but the convergence of the algorithm is only proved in the convex setting. Since the main motivation of the algorithm is finding flat minima, the lack of convergence proof for nonconvex setting concerns me.
>
> A: The motivation of our algorithm is twofold. Finding the minima that generalize better for deep neural nets, and reduce the variance of the stochastic gradient on-the-fly. We provided analysis and experiments to validate the variance reduction and optimality gap reduction for the convex optimization problem. We also provided a large number of experiments to verify the advantages of the proposed algorithm in training deep neural nets.
>
>
> Q3. In the neural net experiment in section 4.1, both gradient descent and smooth gradient descent use the same step size. It is known that the performance of gradient descent is sensitive to the choice of step sizes, for a fair comparison, one should compare the performance of the two algorithms using optimized stepsizes.
>
> A: For experiments in section 4.1, we use the commonly used step size for SGD in training the neural nets. In convex optimization, LS-SGD allows a bigger step size, but in training deep neural nets, it is very hard to determine what is the optimal step size. Indeed, finding a better step size of LSGD when training deep neural nets is still under our investigation.
>
>    However, for convex optimization, we have explored the step size issue at the beginning of Section 3, where for the simple quadratic function, we showed that Laplacian smoothing allows a much bigger time step, which increases the largest time step from 0.9 to 1.8.
>
>
> Q4. In the experiment in section 4.2, the proposed algorithm is only used for the first 40 epochs during training and SGD is used for the later phase of training. Why switching to SGD later?
>
> A: Thanks for your question. Indeed this is motivated by the landscape of ResNet’s loss function (visualization of ResNet’s landscape can be found in the paper: Li et al, arXiv: 1712.09913, 2017). Locally, the loss function of ResNet is much smoother than neural nets without skip connections, i.e., there are not many sharp minima. In training ResNet, we noticed that after 40 epochs, the loss does not change very quickly, and we conjecture it reaches such a smooth region. LS-SGD involves FFT which is slightly slower than SGD, and since we train ResNet56 50 times. To save GPU time, we switched to SGD. We have also tested using LS-SGD for the entire training procedure, The performances are almost identical when switching to SGD after 40 epochs.

---

### Official Review · AnonReviewer3 · 2018-11-02
**Solid algorithm and results, some concerns, tiny font**

**Rating:** 6
**Confidence:** 4

**Review:**

The paper considers SGD with a scaled norm; in the non-stochastic case (first equation in section 2.1), it is gradient descent in a fixed non-Euclidean norm, but it is the stochastic case that is most interesting. The paper connects this, somewhat, to a Hamilton-Jacobi equation, but then relaxes the implicit step to an explicit step.

There is solid theory (Prop 2, 3 and 4) for convergence, which makes sense since this is the same as usual SGD but in a different Hilbert space. Since the inner product is stationary, it's just a fixed Hilbert space, so any convergence proofs that work for arbitrary Hilbert space immediately give the result.

The computational experiments are impressive, and demonstrate a lot of competence with modern neural nets. Some results are hard to interpret (Figs 9, 10) though.

As for why use the Laplacian, Prop 8 (combined with Prop 6) gives some idea: that we lower the variance, without cheating (ie., we could trivially lower the variance by just multiplying by a small number, but because the operator preserves the sum of the components, it is not "cheating").  That is helpful, though it doesn't give a complete picture yet.  The explanation about the link to the "more convex" function I find completely inaccurate and misleading (see technical comments for why).

The writing is mainly fine, though some sentences are written poorly and would benefit from a revision, e.g., 2nd paragraph, "But none of them is suitable to train deep neural nets (DNNs)." is quite awkward [also, in this sentence, please explain *why* they are not suitable!]

The paper circumvents the page limit by using a smaller font (starting on page 3). This might seem like a minor issue, but it is violating the page limit, and not fair to other papers (unless I have misunderstood; the meta-reviewers can probably comment about this).  I do not think it would be unfair to reject the paper on these grounds. It leaves a bad taste in my mouth after reading the paper.


Technical comments:

- page 1, this is called a "tri-diagonal" linear system, but it is not, it is circulant due to the upper-right and lower-left entries (the authors are well-aware of this, but the reader maybe confused; especially since if it were tri-diagonal, it would be inverted via the Thomas algorithm not the FFT).

- Section 2: my first impression on reading this is that you've re-discovered the proximal point envelope and the Moreau envelope (and, looking at the proof of Prop. 1, the authors are aware of this connection).  In this context, it's not clear why A_sigma is helpful, as opposed to any positive definite matrix.

- The actual statement of Proposition 1 is unclear. What does "the ... update ... permits .." mean? i.e., "permits" is a weird, vague choice of words. What are you actually proving?

- Section 2.1 moves from the proximal point method (in a scaled norm) to the gradient descent method (in a scaled norm). Clearly, these two methods are different, and just as in ODE schemes, the implicit version is unconditionally stable while the explicit one isn't. So motivating your method by "smoothing" or "adding convexity" is really misleading. You could define equation (2) and the u(w,t) equation by replacing A_sigma with the identity, and as long as tau > 0, this also "convexifies", but then if you go from implicit to explicit, you get regular GD, so you haven't really done anything.  So, I do not buy this connection that your method "convexifies" the function.

- Using the FFT to invert seems slow (the theoretical flop-count is good, but it's still super-linear, and requires global data movement, so not good for a distributed implementation).  If you really did define A to be tri-diagonal, then you could invert naively in a linear time algorithm with local data movement. Why not use a tri-diagonal A? It might not satisfy prop 8 exactly, but it'd be close, and a lot faster in practice.  From a "finite-difference" point-of-view, I don't see an inherent argument about why you want circular boundary conditions.

- Remark 1 seems out-of-place. Why is that included?

- Section 3.2, and Fig. 5.  It's not clear that the improved generalization results are due to broader local minima, or if it's because the methods converged faster on the training data (since they were limited to 200 epochs). Showing the training error, as a function of epoch, would help clarify. Similar comment for other experiments too.

- Section 4 was impressive in the implementations. Nice work.

- The acknowledgments used the boiler-plate latex template text.

** summary **
Quality: Good
Clarity: OK
Originality: mixed
Significance: maybe high?

---

> ### Author Response · Authors · 2018-11-17
> **Response to Reviewer #3**
>
> We appreciate the reviewer’s support for the influence, solid algorithm and the results of our manuscript. As you will see, the helpful comments lead to a meaningful improvement of our revised manuscript.
>
> Q1. The writing is mainly fine, ..., is quite awkward [also, in this sentence, please explain *why* they are not suitable!]
>
> A: We are sorry for this unsuitable statement. In the revised paper we have changed the statement to “These algorithms have a certain amount of difficulty in applying to train deep neural nets (DNN). SAGA has a relatively high space complexity in storing the gradient for many samples. SVRG requires  computation of the full batch gradient.”
>
> Q2. The paper circumvents the page limit by using a smaller font (starting on page 3). .... It leaves a bad taste in my mouth after reading the paper.
>
> A: Thanks for pointing out this mistake, in the revised manuscript. We have fixed the font size issue.
>
> Q3. page 1, this is called a "tri-diagonal" linear system, but it is not, it is circulant due to the upper-right and lower-left entries.
>
> A: The matrix is a special circulant matrix. We have corrected the original inappropriate terminology in the updated paper.
>
> Q4.  Section 2: my first impression on reading this is that you've re-discovered the proximal point envelope and the Moreau envelope. In this context, it's not clear why A_sigma is helpful, as opposed to any positive definite matrix.
>
> A: The specially designed Moreau envelope serves as our initial motivation for this algorithm. One of the major advantages of A_sigma is that it reduces the variance of the stochastic gradient. We have made this motivation clearer in the revised manuscript.
>
> Q5.  The statement of Proposition 1 is unclear. What does "the ... update ... permits .." mean? i.e., "permits" is a weird, vague choice of words. What are you actually proving?
>
> A: We have changed the statement to “is equivalent to”. In this Proposition, we briefly show the connection between the Moreau envelope and implicit gradient descent (or backward Euler method).
>
> Q6.  Section 2.1 moves from the proximal point method (in a scaled norm) to the gradient descent method (in a scaled norm). .... So, I do not buy this connection that your method "convexifies" the function.
>
> A: You are correct, this is a motivation for our algorithm. In Figure.2, we do see that LS-GD helps to bypass local minima, while GD is trapped into the local minima.
>
> Q7. Using the FFT to invert seems slow (the theoretical flop-count is good, but it's still super-linear and requires global data movement, so not good for a distributed implementation). If you really did define A to be tri-diagonal, then you could invert naively in a linear time algorithm with local data movement. Why not use a tri-diagonal A? It might not satisfy prop 8 exactly, but it'd be close, and a lot faster in practice. From a "finite-difference" point-of-view, I don't see an inherent argument about why you want circular boundary conditions.
>
> A:  Replacing the circulation matrix by the tridiagonal one would degrade the performance of the variance reduction. This would introduce discontinuities in the right hand side, which kills one of the main points of our algorithm. We agree with the point that FFT is super-linear and requires global data movement. However, for the circulant matrix we used, we can adopt the Thomas algorithm together with the Sherman-Morrison formula to invert it in linear time. However, for the high order smoothing schemes, where the matrix becomes (I+(-1)^n\sigma L^n), the FFT algorithm has exactly the same time complexity. Indeed, we are considering improving the FFT component from a parallel and distributed computing point of view.
>
> Q8. Remark 1 seems out-of-place. Why is that included?
>
> A: The Sobolev gradient is a related technique in image processing.
>
> Q9. Section 3.2, and Fig. 5. It's not clear that the improved generalization results are due to broader local minima, or if it's because the methods converged faster on the training data (since they were limited to 200 epochs). Showing the training error, as a function of epoch, would help clarify.
>
> A: Thanks for the constructive comments. In the appendix of the updated paper, Fig.10, we have included the plot training errors that correspond to the results in Fig. 5. Over 100 independent experiments, we do see that all the four algorithms converge when 200 epochs are used. LS-SGD with a different smoothing order really reduces variance compared to SGD. SVRG has nearly insignificant variance over different experiments, but the generalization is not as good as LS-SGD.

---

> > ### Comment · AnonReviewer3 · 2018-11-26
> > **Read authors' response**
> >
> > I've read the authors response. It's a reasonable response, but I don't think my evaluation changes much. I still find the connection with the proximal point version misleading, since we all know that implicit and explicit methods are so different -- I think this could still be mentioned, but should not be a prominent part of the paper (and please don't say things about making the problem "more convex" unless you're trying to take credit for re-discovering the proximal point algorithm).
> >
> > The main contribution is the variance-reduction. As far as I know, this exact type of variance reduction is new, so I think it's an interesting contribution, which is why I still have the vote for "weak accept", but I think the paper would benefit from being reworked, especially the experiments. I'm still impressed by the experiments, but as most reviewers mentioned, there are issues about step-sizes (i.e., trying to be fair for the different methods, since the natural step-size might be different for some methods).

---

> > > ### Author Response · Authors · 2018-11-27
> > > **Thanks for the response**
> > >
> > > We appreciate the reviewer's effort in evaluating our work and propose lots of helpful comments. We will revise our paper follow the above comments.

---

### Public Comment · ~Siwei_Luo1 · 2018-10-09
**Hamilton-Jacobi PDEs and convexification**


It is interesting to see the Hamilton-Jacobi PDEs application in optimization. My concern and curiosity  is that how to make sure that the global minimum remains to be global minimum point during the convexification. In other words, how to prove that this method will not change the geometrical shape of objective function dramatically.

And the relationship between Hamilton-Jacobi PDEs and the operator A_{\sigma} is not quite obvious. Could you please give more instruction and details to demonstrate the motivation of this method?

---

> ### Author Response · Authors · 2018-10-10
> **Reply to "Hamilton-Jacobi PDEs and convexification"**
>
> Thank you for the questions and comments.
> The Hopf-Lax formula for u(w,t), written below equation 2, contains the answer to both questions.
> Let v* be a minimizer of f(v). Then the quantity inside the brackets whose inf over v we are taking is always greater than or equal to f(v), since A_sigma is positive definite. So it is clear that u(w, t) is always >= f(v*), i.e., we set v=w inside the brackets and w=v* is the minimizer of u(w,t). for any t > 0. Next, in the proof of Proposition 1, we derive a formula for the w gradient of u(w, t). gradu(w,t)=-1/tA_sigma(v(w, t)-w)=gradf(v(w, t)).  There is a missing - sign here in our paper. Typo. Sorry!!!
> Let w=w(k), so v(w_k,t)=w(k+1). Then  iterating
> w(k+1)=w(k)-t(A_sigma)^(-1)grad f(w(k+1))  is the same as iterating w(k+1)=w(k)-t grad u(w(k),t)).
>
> The global minimum and relatively flat minimum remains unchanged!

---

> > ### Comment · AnonReviewer3 · 2018-11-02
> > **Proximal Point**
> >
> > Another way to see it is via the usual proximal point characterization:
> >
> > min_w f(w) = min_z [ min_w   f(w) + 1/t ||w-z||^2 ]
> >
> > for any t>0.  To prove this equality, note the LHS <= RHS since the RHS has an extra non-negative term and the same f(w) term. But the LHS = RHS in fact because you can just choose z=w

---

### Public Comment · (anonymous) · 2018-10-21
**Question About The selection of A**

I wonder that what is the benefit of choosing u_t+<u_x,Au_x> =0 rather than the deep relaxation PDE u_t+<u_x,u_x>=0

It seems that the paper doesn't mention the reason that why A here should be the laplacian matrix.

p.s. It seems that the choice of A isn't invariant to the perturbation of coordinate

---

> ### Author Response · Authors · 2018-10-21
> **Reply to "Question About The Selection of A"**
>
> Thank you for the questions and comments.
>
> Laplacian matrix has advantages in reducing noise in stochastic gradient descent. When sigma=0, it recovers the deep relaxation.
>
> Our numerical results demonstrate that the coordinate perturbation has minimal influence on the results. We welcome readers to test our schemes and further advances them together.

---

> ### Comment · AnonReviewer3 · 2018-11-02
> **Agreed**
>
> In the non-stochastic case, I agree with you 100%: there is no clearly shown benefit to using A instead of the identity (i.e. sigma=0 case).  Their motivation for using A is for the stochastic case, see Prop 8.  But I agree, the connection to the HJ-PDE does not depend on the explicit form of A (as long as it is positive definite).

---

> > ### Public Comment · (anonymous) · 2018-12-31
> > **Some concern**
> >
> > But I think any A if A performs like an averaging operator can dones this, their proof seems that like averaging "independent noise" can reduce the variance.... However I don't think this is want happens in the real sgd.

---

### Meta-Review · Area_Chair1 · 2018-12-14
**Issues with the experiments**

**Confidence:** 4
**Recommendation:** Reject

**Metareview:**

Dear authors,

The topic of variance reduction in optimization is timely and the reviewers appreciated your attempt at circumventing the issues faced with the current popular methods.

They however had a concern about the significance of the results, which I echo:
- First, there have been previous attempts at variance reduction which share some similarity with yours, for instance "No more pesky learning rate", "Topmoumoute online natural gradient algorithm" or even Adam (which does variance reduction without mentioning it).
- The fact that previous similar methods exist is a non-issue should yours perform better. However, the absence of stepsize tuning in the experimental evaluation is a big issue as the performance of an iterative algorithm is highly sensitive to it.

Finally, the link between flatness of the minimum and generalization is dubious, as mentioned for instance by Dinh et al. (2017).

As a consequence, I cannot accept this work for publication to ICLR but I encourage you to address the points of the reviewers should you wish to resubmit it to a future conference.